# BIAS AMPLIFICATION IMPROVES WORST-GROUP ACCURACY WITHOUT GROUP INFORMATION

## ABSTRACT

Neural networks produced by standard training are known to suffer from poor accuracy on rare subgroups despite achieving high accuracy on average, due to the correlations between certain spurious features and labels. Previous approaches based on worst-group loss minimization (*e.g.* Group-DRO) are effective in improving worse-group accuracy but require expensive group annotations for all the training samples. In this paper, we focus on the more challenging and realistic setting where group annotations are only available on a small validation set or are not available at all. We propose BAM, a novel two-stage training algorithm: in the first stage, the model is trained using a *bias amplification* scheme via introducing a learnable *auxiliary variable* for each training sample together with the adoption of squared loss; in the second stage, we upweight the samples that the bias-amplified model misclassifies, and then continue training the same model on the reweighted dataset. Empirically, BAM leads to consistent improvement over its counterparts in worst-group accuracy, resulting in state-of-the-art performance in spurious correlation benchmarks in computer vision and natural language processing. Moreover, we find a simple stopping criterion that completely removes the need for group annotations, with little or no loss in worst-group accuracy.

## 1 INTRODUCTION

The presence of spurious correlations in the data distribution, also referred to as "shortcuts" (Geirhos et al., 2020), is known to cause machine learning models to generate unintended decision rules that rely on spurious features. For example, image classifiers can largely use background instead of the intended combination of object features to make predictions (Beery et al., 2018). Similar phenomenon is also prevalent in natural language processing (Gururangan et al., 2018) and reinforcement learning (Lehman et al., 2020). In this paper, we focus on the *group robustness* formulation of such problems (Sagawa et al., 2019), where we assume the existence of *spurious attributes* in the training data and define *groups* to be the combination of class labels and spurious attributes. The objective is to achieve high *worst-group accuracy* on test data, which would indicate that the model is not exploiting the spurious attributes.

Under this setup, one type of method uses a distributionally robust optimization framework to directly minimize the worst-group training loss (Sagawa et al., 2019). While these methods are effective in improving worst-group accuracy, they require knowing the group annotations for all training examples, which is expensive and oftentimes unrealistic. In order to resolve this issue, a line of recent work focused on designing methods that do not require group annotations for the training data, but need them for a small set of validation data (Liu et al., 2021; Nam et al., 2020; 2022; Zhang et al., 2022). A common feature shared by these methods is that they all consist of training two models: the first model is trained using plain empirical risk minimization (ERM) and is intended to be "biased" toward certain groups; then, certain results from the first model are utilized to train a debiased second model to achieve better worst-group performance. For instance, a representative method is JTT (Liu et al., 2021), which, after training the first model using ERM for a few epochs, trains the second model while upweighting the training examples incorrectly classified by the first model.

The core question that motivates this paper is: *Since the first model is intended to be biased, can we amplify its bias in order to improve the final group robustness?* Intuitively, a bias-amplified first model can provide better information to guide the second model to be debiased, which can

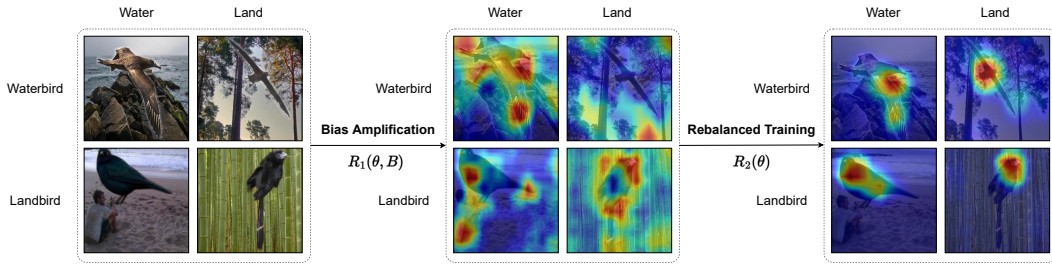

Figure 1: Using Grad-CAM (Selvaraju et al., 2017) to visualize the effect of bias amplification and rebalanced training stages, where the classifier heavily relies on the background information to make predictions after bias amplification but focuses on the useful feature (bird) itself after the rebalanced training stage.

potentially lead to improving group robustness. To this end, we propose a new two-stage algorithm, BAM (Bias AMplification), for improving worst-group accuracy without any group annotations for training data:

- *Stage 1: Bias amplification.* We train a bias-amplified model by introducing a trainable auxiliary variable for each training example, as well as using squared loss instead of cross-entropy loss.

- *Stage 2: Rebalanced training.* We upweight the training examples that are misclassified in Stage 1, and continue training the same model instead of retraining a new model.[1]

Evaluated on various benchmark datasets for spurious correlations, including Waterbirds (Wah et al., 2011; Sagawa et al., 2019), CelebA (Liu et al., 2015; Sagawa et al., 2019), MultiNLI (Williams et al., 2018; Sagawa et al., 2019), and CivilComments-WILDS (Borkan et al., 2019; Koh et al., 2021), we find that BAM achieves state-of-the-art worst-group accuracy compared to existing methods that only use group annotations on a validation set for hyperparameter tuning. We also conduct a detailed ablation study and observe that every component in BAM (auxiliary variables, squared loss, continued training) is crucial in its improved performance.

Furthermore, we explore the possibility of *completely* removing the need for group annotations. We find that low-class accuracy difference (which does not require any group annotations to evaluate) is strongly correlated with high worst-group accuracy. Using minimum class accuracy difference as the stopping criterion, BAM outperforms the previous state-of-the-art annotation-free method, GEORGE (Sohoni et al., 2020), by a considerable margin and closes the performance gap between GEORGE and fully-supervised Group-DRO by an average of 88% on the image classification datasets.

## 2 RELATED WORKS

A variety of recent work discussed different realms of robustness, for instance, class imbalance (He & Garcia, 2009; Huang et al., 2016; Khan et al., 2017; Johnson & Khoshgoftaar, 2019; Thabtah et al., 2020), and robustness in distribution shift, where the target data distribution is different from the source data distribution (Clark et al., 2019; Zhang et al., 2020; Marklund et al., 2020; Lee et al., 2022; Yao et al., 2022). In this paper, we mainly focus on improving group robustness. Categorized by the amount of information we have for training and validation, we discuss three directions below:

**Improving Group Robustness with Training Group Annotations.** Multiple works have used training group annotations to improve worst-group accuracy (Byrd & Lipton, 2019; Khani et al., 2019; Goel et al., 2020; Cao et al., 2020; Sagawa et al., 2020). Other works include minimizing the worst-group training loss using distributionally robust optimization (Group-DRO) (Sagawa et al., 2019), simple training data balancing (SUBG) (Idrissi et al., 2022), and retraining the last layer of the

---

[1]In Figure 1, we use Grad-CAM visualization to illustrate that our bias-amplified model from Stage 1 focuses more on the image background while the final model after Stage 2 focuses on the object target.

model on the group-balanced dataset (DFR) (Kirichenko et al., 2022). These methods achieve state-of-the-art performance on all benchmark datasets. However, the acquisition of spurious attributes of the entire training set is extremely expensive and unrealistic in real-world datasets.

**Improving Group Robustness with Validation Group Annotations Only.** Acknowledging the cost of obtaining group annotations, many recent works focus on the setting where training group annotations are not available (Duchi & Namkoong, 2019; Oren et al., 2019; Levy et al., 2020; Pezeshki et al., 2021). Taghanaki et al. (2021) proposes a transformation network to remove the spurious correlated features from image datasets and then choose classifier architectures according to the downstream task. Shu et al. (2019) utilizes a small set of unbiased meta-data to reweight data samples. CVaR DRO (Duchi et al., 2019) introduces a variant of distributionally robust optimization that dynamically reweights data samples that have the highest losses. In particular, the most popular recent methods of achieving high worst-group accuracies involve training two models. CNC (Zhang et al., 2022) first trains an ERM model to help infer pseudo group labels by clustering output features and then adopts a standard contrastive learning approach to improve robustness. SSA (Nam et al., 2022) uses a subset of the validation samples with group annotations for training to obtain pseudo spurious attributes, and then trains a robust model by minimizing worst-group loss, the same as Group-DRO. Similarly, $\text{DFR}_{\text{Tr}}^{\text{Val}}$ (Kirichenko et al., 2022) uses validation data with group annotations for training and tuning hyperparameters, though it just requires retraining the last layer of the model.

Approaches that are related to our method normally use the first model to identify minority samples and then train a separate model based on the results predicted by the first model (Yaghoobzadeh et al., 2019; Utama et al., 2020). LfF (Nam et al., 2020) train two models concurrently, where one model is intentionally biased, and the other one is debiased by reweighting the gradient of the loss according to a relative difficulty score. JTT (Liu et al., 2021) first trains an ERM model to identify minority groups in the training set (similar to EIIL (Creager et al., 2021)), and then trains a second ERM model with these selected samples to be upweighted. However, the above two-model approaches all focus on the robust training of the second model and fail to consider the potential of accumulating biased knowledge from the first model. Kim et al. (2022) introduces a multi-model approach that proposes to identify hard-to-learn samples and obtain their weights based on consensus of member classifiers of a committee, and simultaneously train a main classifier through knowledge distillation. In addition, despite the reduced reliance, all these approaches still require a small amount of group-annotated samples.

**Improving Group Robustness without any Group Annotations.** Relatively little work has been done under the condition that no group information is provided for both training and validation. Idrissi et al. (2022); Liu et al. (2021) observe a significant drop (10% - 25%) in worst-group test accuracy if using the highest *average* validation accuracy as the stopping criterion without any group annotations. One of the recent works, GEORGE (Sohoni et al., 2020), tries to separate unlabeled classes in deep model feature spaces and then use the generated pseudo labels to train the model via the distributionally robust optimization objective. Additionally, Seo et al. (2022) clusters the pseudo-attributes based on the embedding feature of a naively trained model, and then define a trainable factor to reweight different clusters based on their sizes and target losses. However, there is a considerable performance gap between the unsupervised and the supervised methods.

## 3 PRELIMINARIES

We adopt the group robustness formulation for spurious correlation (Sagawa et al., 2019). Consider a classification problem where each sample consists of an input $x \in \mathcal{X}$, a label $y \in \mathcal{Y}$, and a spurious attribute $a \in \mathcal{A}$. For example, in CelebA, $\mathcal{X}$ contains images of human faces and we want to classify hair color into the labels $\mathcal{Y} = \{\text{blonde}, \text{not blonde}\}$. Hair color may be highly correlated with gender $\mathcal{A} = \{\text{male}, \text{female}\}$ which is a spurious feature that can also predict the label. We say that each example belongs to a group $g = (y, a) \in \mathcal{G} = \mathcal{Y} \times \mathcal{A}$.

Let $f : \mathcal{X} \to \mathcal{Y}$ be a classifier learned from a training dataset $D = \{(x_i, y_i)\}_{i=1}^n$. We hope that $f$ does not overly rely on the spurious feature $a$ to predict the label. To this end, we evaluate the model through its *worst-group error*:

$$\text{Err}_{\text{wg}}(f) := \max_{g \in \mathcal{G}} \mathbb{E}_{x, y|g}[\mathbb{1}[f(x) \neq y]].$$

We focus on the setting where no group annotations are available in the training dataset. We consider two cases under this setting: (1) group annotations are available in a validation set solely for the purpose of hyperparameter tuning, and (2) no group annotations are available at all. We will distinguish between these cases when comparing them with existing methods.

# 4 OUR APPROACH: BAM

---
**Algorithm 1** BAM

---
**Input:** Training dataset $D$, number of epochs $T$ in Stage 1, auxiliary coefficient $\lambda$, and upweight factor $\mu$.

 **Stage 1: Bias Amplification**

 1. Optimize $R_1(\theta, B)$ (1) for $T$ epochs and save the model parameters $\hat{\theta}_{\text{bias}}$.

 2. Construct the error set $E$ (2) misclassified by $\hat{f}_{\text{bias}}(\cdot) = f_{\hat{\theta}_{\text{bias}}}(\cdot)$.

 **Stage 2: Rebalanced Training**

 3. Continue training the model starting from $\hat{\theta}_{\text{bias}}$ to optimize $R_2(\theta)$ (3).

 4. Apply a stopping criterion:

  • If group annotations are available for validation, stop at the highest worst-group validation accuracy;

  • If no group annotations are available, stop at the lowest validation class difference (4).

---

We now present BAM, a two-stage approach to improving worst-group accuracy without any group annotations at training time. In Stage 1, we train a *bias-amplified model* and select examples that this model makes mistakes on. Then, in Stage 2, we continue to train the same model while upweighting the samples selected from Stage 1.

## 4.1 STAGE 1: BIAS AMPLIFICATION

The key intuition behind previous two-stage approaches (*e.g.* JTT) is that standard training via ERM tends to first fit easy-to-learn groups with spurious correlations, but not the other hard-to-learn groups where spurious correlations are not present. Therefore, the samples that the model misclassified in the first stage can be treated as a proxy for hard-to-learn groups and used to guide the second stage.

We design a bias-amplifying scheme in Stage 1 with the aim of identifying a higher-quality error set to guide training in Stage 2. In particular, we introduce a trainable auxiliary variable for each example and add it to the output of the network. Let $f_\theta : \mathcal{X} \to \mathbb{R}^C$ be the neural network with parameters $\theta$, where $C = |\mathcal{Y}|$ is the total number of classes. We use the following objective function in Stage 1:

$$R_1(\theta, B) = \frac{1}{n} \sum_{i=1}^{n} \ell(f_\theta(x_i) + \lambda b_i, y_i). \tag{1}$$

Here, $b_i \in \mathbb{R}^C$ is the auxiliary variable for the $i$-th example in the training set, and the collection of auxiliary variables $B = (b_1, \ldots, b_n)$ is learnable and is learned together with the network parameters $\theta$ ($B$ is initialized to be all 0). $\lambda$ is a hyperparameter that controls the strength of the auxiliary variables. We adopt the squared loss $\ell(z, y) = \|z - e_y\|_2^2$ where $e_y \in \mathbb{R}^C$ is the one-hot encoding for the label $y$. Below we explain the main ideas behind our method.

The introduction of auxiliary variables makes it more difficult for the network $f_\theta$ to learn, because the auxiliary variables can do the job of fitting the labels. We expect this effect to be more pronounced for hard-to-learn examples. For example, if in normal ERM training it takes a long time for the network $f_\theta$ to fit a hard-to-learn example $(x_i, y_i)$, after introducing the auxiliary variable $b_i$, it will be much easier to use $b_i$ to fit the label $y_i$, thus making the loss $\ell(f_\theta(x_i) + \lambda b_i, y_i)$ drop relatively faster. This will prohibit the network $f_\theta$ itself from learning this example. The such effect will be smaller for easy-to-learn examples, since the network itself can still quickly fit the labels without much reliance on the auxiliary variables. Therefore, **adding auxiliary variables amplifies the bias toward easy-to-learn examples**, making hard examples even harder to learn.

We note that the auxiliary variable was first introduced in Hu et al. (2020) for a different motivation (learning with noisy labels). Hu et al. (2020) proved that it recovers kernel ridge regression when using the squared loss and when the neural network is in the Neural Tangent Kernel (Jacot et al., 2018) regime, and provided a theoretical guarantee for the noisy label learning problem. Empirically, Hu et al. (2020) showed that squared loss works better than cross-entropy loss, which is why we use the squared loss in (1). Additional evidence that squared loss can achieve competitive performance for classification was summarized in Hui & Belkin (2020).

At the end of Stage 1, we evaluate the obtained model $\hat{f}_{\text{bias}}(\cdot) = f_{\hat{\theta}_{\text{bias}}}(\cdot)$ on the training set and identify an *error set*: (note that auxiliary variables are now removed)

$$E = \{(x_i, y_i) \colon \hat{f}_{\text{bias}}(x_i) \neq y_i\}. \tag{2}$$

## 4.2 STAGE 2: REBALANCED TRAINING

In Stage 2, we continue training the model starting from the parameters $\hat{\theta}_{\text{bias}}$ from Stage 1, using a rebalanced loss that upweights the examples in the error set $E$:

$$R_2(\theta) = \mu \sum_{(x,y) \in E} \ell_{\text{CE}}(f_\theta(x), y) + \sum_{(x,y) \in D \setminus E} \ell_{\text{CE}}(f_\theta(x), y), \tag{3}$$

where $\ell_{\text{CE}}$ is the cross-entropy loss and $\mu$ is a hyperparamter (upweight factor).

We note that more complicated approaches have been proposed for Stage 2, *e.g.* Zhang et al. (2022), but we stick with the simple rebalanced training method in order to focus on the bias amplification effect in Stage 1.

## 4.3 STOPPING CRITERION WITHOUT ANY GROUP ANNOTATIONS – CLASS DIFFERENCE

When group annotations are available in a validation set, we can simply use the *worst-group validation accuracy* as a stopping criterion and to tune hyperparameters, similar to prior approaches (Nam et al., 2020; Liu et al., 2021; Creager et al., 2021; Zhang et al., 2022). When no group annotations are available, a naive approach is to use the validation average accuracy as a proxy, but this results in poor worst-group accuracy (Liu et al., 2021; Idrissi et al., 2022).

We identify a simple heuristic when no group annotations are available, using *minimum class difference*, which we find to be highly effective and result in little or no loss in worst-group accuracy. For a classification problem with $C$ classes, we calculate the sum of pairwise validation accuracy differences between classes as

$$\text{ClassDiff} = \sum_{i,j=1}^{C} |\text{Acc}(\text{class } i) - \text{Acc}(\text{class } j)|. \tag{4}$$

ClassDiff can be calculated on a validation set without any group annotations. In all the datasets (where $C = 2$ or $3$) we experiment with, we observe that ClassDiff inversely correlates with worst-group accuracy (see Section 5.4). Therefore, we can use ClassDiff as a stopping criterion and completely remove the need for any group annotations.

Our algorithm is summarized in Algorithm 1. It has three hyperparameters: $T$ (number of epochs in Stage 1), $\lambda$ (auxiliary variable coefficient), and $\mu$ (upweight factor). We provide full training details in Appendix B.

## 5 EXPERIMENTS

In this section, we first briefly explain the experiment setup (Section 5.1). Next, we present our main results and show that BAM improves worst-group accuracy compared to prior methods that are trained without spurious attributes on the same benchmark datasets. BAM achieves state-of-the-art performance when group annotations are either available or unavailable in the validation set (Section 5.2). Then, we conduct thorough ablation studies to verify the effectiveness of every component of our proposed model (Section 5.3). Finally, we provide more detailed analyses of BAM's behavior (Section 5.4).

## 5.1 Setup

We conduct our experiments on four popular benchmark datasets containing spurious correlations. Two of them are image datasets: Waterbirds and CelebA, and the other two are NLP datasets: MultiNLI and CivilComments-WILDS. The full dataset details are in Appendix A. BAM is trained in the absence of training group annotations throughout all experiments. We obtain the main results of BAM via tuning with and without group annotations on the validation set, following Algorithm 1.

For a fair comparison, we adopt the general settings from previous methods (JTT) and stay consistent with other approaches without extensive hyperparameter tuning (batch size, learning rate, regularization strength in Stage 2, etc.). We use pretrained ResNet-50 (He et al., 2016) for image datasets, and pretrained BERT (Devlin et al., 2019) for NLP datasets. More details can be found in Appendix B.

Table 1: Average and worst-group test accuracies of different approaches evaluated on image datasets (Waterbird and CelebA). We run BAM and JTT (in *) on 3 random seeds based on the highest worst-group validation accuracies and minimum class differences, respectively, and report the mean and standard deviation. Results of EIIL and Group-DRO are reported by Nam et al. (2022), and results of other approaches come from their original papers. The best worst-group accuracies under the same condition are marked in **bold**.

| Annotation free? | Annotation only used for tuning h-params? | Method | Waterbird | | CelebA | |
|---|---|---|---|---|---|---|
| | | | Avg. | Worst-group | Avg. | Worst-group |
| No | No | SUBG (Idrissi et al., 2022) | - | $89.1_{\pm1.1}$ | - | $85.6_{\pm2.3}$ |
| | | GroupDRO (Sagawa et al., 2019) | $91.8_{\pm0.48}$ | $\mathbf{89.2}_{\pm0.18}$ | $93.1_{\pm0.21}$ | $88.5_{\pm1.16}$ |
| | | SSA (Nam et al., 2022) | $92.2_{\pm0.87}$ | $89.0_{\pm0.55}$ | $92.8_{\pm0.11}$ | $\mathbf{89.8}_{\pm1.28}$ |
| | Yes | ERM | $97.3$ | $72.6$ | $95.6$ | $47.2$ |
| | | EIIL (Creager et al., 2021) | $96.9$ | $78.7$ | $91.9$ | $83.3$ |
| | | CNC (Zhang et al., 2022) | $90.9_{\pm0.1}$ | $88.5_{\pm0.3}$ | $89.9_{\pm0.5}$ | $\mathbf{88.8}_{\pm0.9}$ |
| | | JTT* | $89.9_{\pm0.41}$ | $86.8_{\pm1.61}$ | $91.3_{\pm0.36}$ | $78.7_{\pm1.15}$ |
| | | BAM | $91.4_{\pm0.44}$ | $\mathbf{89.7}_{\pm0.26}$ | $88.0_{\pm0.37}$ | $83.5_{\pm0.94}$ |
| Yes | - | GEORGE (Sohoni et al., 2020) | $95.7$ | $76.2$ | $94.8$ | $52.4$ |
| | | JTT + ClassDiff* | $88.5_{\pm1.47}$ | $87.1_{\pm0.24}$ | $91.8_{\pm0.76}$ | $75.4_{\pm3.28}$ |
| | | BAM + ClassDiff | $91.4_{\pm0.31}$ | $\mathbf{89.2}_{\pm0.15}$ | $88.4_{\pm2.32}$ | $\mathbf{80.3}_{\pm3.32}$ |

Table 2: Average and worst-group test accuracies of different approaches evaluated on natural language datasets (MultiNIL and CivilComments-WILDS), following the same conventions in Table 1.

| Annotation free? | Annotation only used for tuning h-params? | Method | MultiNLI | | CivilComments-WILDS | |
|---|---|---|---|---|---|---|
| | | | Avg. | Worst-group | Avg. | Worst-group |
| No | No | SUBG (Idrissi et al., 2022) | - | $68.9_{\pm0.8}$ | - | $\mathbf{71.8}_{\pm0.4}$ |
| | | GroupDRO (Sagawa et al., 2019) | $81.4_{\pm1.40}$ | $\mathbf{76.6}_{\pm0.41}$ | $87.7_{\pm1.35}$ | $69.1_{\pm1.53}$ |
| | | SSA (Nam et al., 2022) | $79.9_{\pm0.87}$ | $76.6_{\pm0.66}$ | $88.2_{\pm1.95}$ | $69.9_{\pm2.02}$ |
| | Yes | ERM | $82.4$ | $67.9$ | $92.6$ | $57.4$ |
| | | EIIL (Creager et al., 2021) | $79.4$ | $70.9$ | $90.5$ | $67.0$ |
| | | CNC (Zhang et al., 2022) | - | - | $81.7_{\pm0.5}$ | $68.9_{\pm2.1}$ |
| | | JTT* | $80.0_{\pm0.41}$ | $68.1_{\pm0.90}$ | $87.2_{\pm1.65}$ | $77.7_{\pm1.70}$ |
| | | BAM | $79.6_{\pm1.11}$ | $\mathbf{71.5}_{\pm1.56}$ | $88.3_{\pm0.76}$ | $\mathbf{79.3}_{\pm2.69}$ |
| Yes | - | GEORGE (Sohoni et al., 2020) | - | - | - | - |
| | | JTT + ClassDiff* | $81.2_{\pm0.56}$ | $66.5_{\pm0.56}$ | $87.2_{\pm1.65}$ | $77.7_{\pm1.70}$ |
| | | BAM + ClassDiff | $80.3_{\pm0.99}$ | $\mathbf{71.2}_{\pm1.52}$ | $88.3_{\pm0.76}$ | $\mathbf{79.3}_{\pm2.69}$ |

## 5.2 Main results

Tables 1 and 2 report the average and worst-group test accuracies of BAM and compare it against standard ERM and recently proposed methods under different conditions, including SUBG (Idrissi et al., 2022), JTT (Liu et al., 2021), SSA (Nam et al., 2022), CNC (Zhang et al., 2022), GEORGE

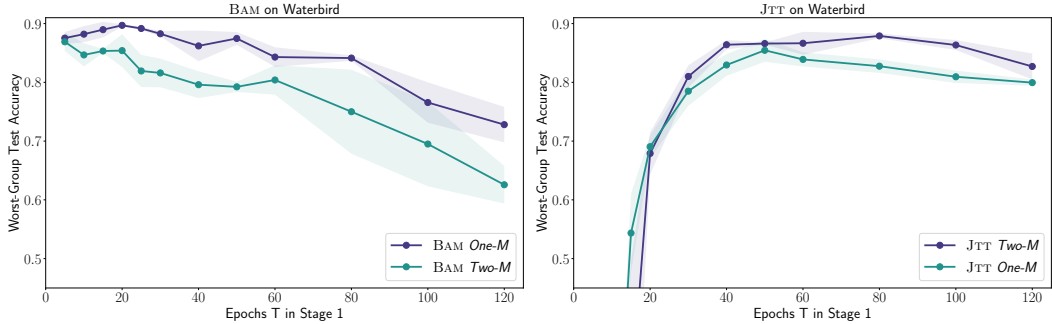

Figure 2: (1) Comparison of BAM's performance between training *One-M* and *Two-M*. (2) Comparison of JTT's performance between training *One-M* and *Two-M*. We find that *One-M* consistently achieves better worst-group test accuracies than *Two-M* for BAM over all hyperparameter combinations, while we do not observe the same result on JTT.

(Sohoni et al., 2020), and Group-DRO (Sagawa et al., 2019). We tune BAM and JTT according to the highest worst-group validation accuracy (not annotation-free) and minimum class difference (annotation-free).

First, compared with other methods that use group annotations only for hyperparameter tuning, BAM consistently achieves higher worst-group accuracies across all datasets, with the exception of the CelebA dataset on which CNC achieves better performance. We note that CNC primarily focuses on improving Stage 2 with a more complicated contrastive learning method, while BAM uses the simple upweighting method. It is possible that the combination of CNC and BAM could lead to better results. Nevertheless, the result of BAM is promising on all other datasets, even surpassing the weakly-supervised method SSA and the fully-supervised method Group-DRO on Waterbirds and CivilComments-WILDS.

Second, if the validation group annotations are not available at all, BAM achieves state-of-the-art performance on all four benchmark datasets and improves previous methods by a large margin. Notably, BAM recovers a significant portion of the gap in worst-group accuracy between GEORGE (previous state-of-the-art that requires no group annotations) and Group-DRO/SSA (previous state-of-the-art requiring supervision) by an average of 88% on the image classification datasets.

BAM's improved performance in worst-group accuracy comes at the expense of a moderate drop in average accuracy. The tradeoff between average accuracy and worst-group accuracy is consistent with the observation made by Liu et al. (2021); Sagawa et al. (2019). We note that our implementation of JTT follows directly from its published code, and we obtain a much higher performance on the CivilComments-WILDS dataset than originally reported by Liu et al. (2021).

## 5.3 ABLATION STUDIES

We conduct thorough ablation studies on the Waterbirds dataset to test the effectiveness of every component of our model (the use of auxiliary variable and the squared loss in Stage 1; continued training in Stage 2). For consistent terminology, we define the approach that loads the model from Stage 1 and continues training the same model in Stage 2 as *One-M*. We define the approach that trains a separate ERM model in Stage 2 as *Two-M*. In this subsection, we first verify the effectiveness of using *One-M* in Stage 2. Then, we study the necessity of using the combination of the auxiliary variable and the squared loss in Stage 1. We have a total of 8 ablations: Train *One-M* using (1) cross-entropy loss, (2) squared loss, (3) auxiliary variable (AUX.), and cross-entropy loss, (4) auxiliary variable and squared loss in Stage 1; Train *Two-M* using (5) cross-entropy loss, (6) squared loss, (7) auxiliary variable and cross-entropy loss, (8) auxiliary variable and squared loss in Stage 1. We note that (1) & (5), (2) & (6), (3) & (7), and (4) & (8) share the same model in Stage 1 and have the same error set to be upweighted. Notably, (4) recovers the standard procedure of BAM and (5) recovers the standard procedure of JTT. For a fair comparison, we employ the same hyperparameters throughout our ablation studies. Stage 2 of (1) - (8) are tuned with identical procedures. We use the same stopping criterion (highest worst-group validation accuracy) for ablation studies. We tune

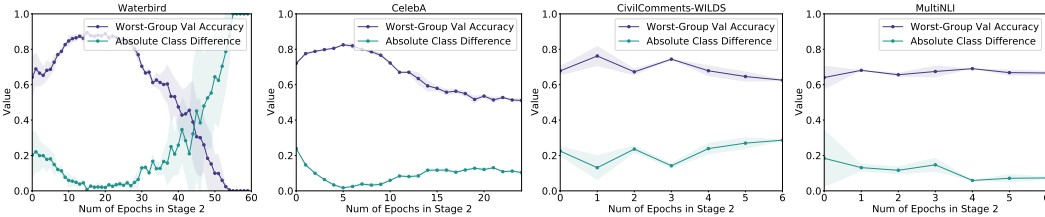

Figure 3: Relation between absolute valdiation class difference and worst-group validation accuracy in Stage 2 on Waterbirds, CelebA, CivilComments-WILDS, and MultiNLI. It can be observed that minimizing absolute validation class difference is roughly equivalent to maximizing worst-group accuracy in the validation set. Each dataset uses the same hyperparameters as employed in Table 1 and Table 2. Each line represents the value averaged over 3 different seeds and the shade represents the standard deviation.

over $T = \{10, 15, 20, 25, 30, 40, 50, 60, 80, 100, 120\}$ and $\mu = \{50, 100, 140\}$ for each setting, and fix $\lambda = 0.5$ whenever auxiliary variables are involved. To reduce the influence of randomness, we run all experiments using three different seeds and report the mean of the worst-group accuracies in Tables 3 and 4. By analyzing the results, we have the following observations:

**_One-M_ is better than _Two-M_ only after bias amplification**   We compare highest worst-group accuracies between _One-M_ and _Two-M_ approaches under all different combinations of components in Stage 1: the use of auxiliary variable and the choice of loss functions. Namely, we compare the performance of the model (1) vs. (5), (2) vs. (6), (3) vs. (7), and (4) vs. (8), as shown in Table 3. The experiment result shows that with the same error set and hyperparameters and using our proposed elements, continuously training one model consistently achieves higher worst-group accuracies, averaging a performance gain of 2.3%. However, _One-M_ itself does not provide improvement for JTT, as shown in (1) vs. (5).

Table 3: Ablation on _One-M_ versus _Two-M_

| Ablation | Worst-Group Acc.(%) |
|---|---|
| (1) _One-M_, Cross-Entropy Loss | 86.4 |
| (5) _Two-M_, Cross-Entropy Loss | 86.8 (**+0.4**) |
| (2) _One-M_, Aux. Variable and Cross-Entropy Loss | 88.7 |
| (6) _Two-M_, Aux. Variable and Cross-Entropy Loss | 86.7 (**-2.0**) |
| (3) _One-M_, Squared Loss | 88.8 |
| (7) _Two-M_, Squared Loss | 86.6 (**-2.2**) |
| (4) _One-M_, Aux. Variable and Squared Loss | 89.7 |
| (8) _Two-M_, Aux. Variable and Squared Loss | 87.0 (**-2.7**) |

Figure 2 compares the highest worst-group test accuracy between _One-M_ BAM/JTT and _Two-M_ BAM/JTT over a wide range of Stage 1 epochs $T$. The result suggests that _One-M_ BAM outperforms _Two-M_ BAM in every single Stage 1 epoch $T$, while such findings do not apply to JTT. In other words, _One-M_ is better than _Two-M_ only after BAM's bias amplification. In particular, BAM outperforms JTT with much fewer epochs $T$ needed in Stage 1, and the _Two-M_ BAM has slightly better worst-group performance than JTT. Furthermore, similar to Liu et al. (2021), the worst-group test accuracy of BAM only stays high for a range of epochs $T$ (10 - 25) in Stage 1 and degrades as $T$ gets large.

**Every component in Stage 1 helps with improving group robustness** After illustrating the superior performance of _One-M_, here we systematically investigate the necessity of the use of our proposed components in Stage 1, namely the combination of the auxiliary variable and the loss function. In Table 4, we apply _One-M_ approach with the same fair setup in all four experiments in Stage 1. We consider the sole use of cross-entropy loss in Stage 1 as our baseline model.

Table 4: Ablation on auxiliary variable and loss functions

| Ablation | Worst-group Acc.(%) |
|---|---|
| (1) Baseline (Cross-Entropy Loss) | 86.4 |
| (2) Baseline + Aux. Variable | 88.7 (**+2.3**) |
| (3) Baseline + Squared Loss | 88.8 (**+2.4**) |
| (4) Baseline + Aux. Variable + Squared Loss | 89.7 (**+3.3**) |

The addition of the auxiliary variable and the substitution of the loss function is indicated with the sign "+". The results suggest that the isolated use of the auxiliary variable, squared loss and the combination of both all contribute to non-trivial improvements in performances over the baseline

model. In particular, the use of squared loss biases the model faster in Stage 1, achieving its best performance with only $T = 20$ epochs compared to JTT's 60 epochs in Stage 1, as illustrated by Figure 2.

## 5.4 FURTHER ANALYSES

**Relation between class difference and worst-group accuracy** Figure 3 plots the trend of the absolute class difference and the worst-group accuracy on the validation set in Stage 2 for the Waterbirds, CelebA, MultiNLI and CiviComments datasets. Clearly, there is a strong inverse relationship between the absolute class difference and worst-group accuracy in the validation set on all four datasets, which justifies the use of class difference as a stopping criterion.

Table 5: Error set specific information. The statistics directly follow from our ablation studies in Section 5.3. AUX or SQ respectively represents the sole use of auxiliary variable or squared loss in Stage 1. Each entry is evaluated by calculating the mean of the error sets from three different seeds. The last row indicates the best worst-group test accuracy evaluated using *One-M* and *Two-M* approaches. It is reasonable to list them together because the one/two model ablation uses the identical error set. We also experiment to train a model that only upweights the "minority examples" (waterbirds in land and landbirds in water) according to our understanding.

| Group Size | Group Description | JTT | AUX | SQ | BAM | Minority-Only |
|---|---|---|---|---|---|---|
| 3498 | Waterbird in Water | 5 | 8 | 17 | 12 | 0 |
| 184 | Waterbird in Land | 63 | 76 | 85 | 81 | 184 |
| 56 | Landbird in Water | 47 | 46 | 36 | 35 | 56 |
| 1057 | Landbird in Land | 114 | 99 | 96 | 91 | 0 |
| -/- | Worst-group Accuracy (*One-M*/*Two-M*) | 86.4/86.8 | 88.7/86.7 | 88.8/86.6 | 89.7/87.0 | 79.2 |

**Error Set Related Analysis** In Table 5, all listed methods identify a large portion of minority examples. After using our proposed components, the error set indicates that the classifier becomes more "biased" towards the majority groups. This is manifested by the amount of majority and minority examples misclassified by the classifier in Stage 1. BAM, AUX, and SQ all contribute to more minority group examples (# waterbird in land + # landbird in water) and fewer majority group examples (# waterbird in water + # landbird in land) misclassified compared with JTT. However, we note that the combination of squared loss and auxiliary variable (BAM) does not guarantee a linearly more "biased" error set compared to their isolated usages. Nevertheless, a seemingly more "biased" error set does not necessarily guarantee optimal performance. The extreme condition, where only the minority samples are upweighted, contributes to a worst-group test accuracy that is substantially lower than every method listed in Table 5.

## 6 DISCUSSION

In this paper, we present BAM, a two-stage method that does not use any training group annotations and achieves state-of-the-art worst-group performance. We also introduce a novel stopping criterion that completely removes the need for any group annotations and is applicable to other methods as well. We conclude our paper by proposing several directions for future work:

First, it is crucial to have a better theoretical understanding of our "bias amplification" scheme. While we conduct thorough ablation studies and provide empirical evidence to support our proposed method, the definition of "bias" and its effects on the model still needs to be understood theoretically. It is also helpful to systematically investigate the influence of the loss functions.

Second, aside from the success on the four benchmark datasets, it would be interesting to see if the class difference can be applied to other datasets, especially those with much more classes.

Finally, while we use simple upweighting in Stage 2, it is possible to combine our approach in Stage 1 with other recently developed methods for Stage 2, such as CNC (Zhang et al., 2022). Any future work that focuses on improving Stage 2 could also use our general framework.

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

# A  DATASET DETAILS

**Waterbirds** (Wah et al., 2011; Sagawa et al., 2019): Waterbirds is an image dataset consisting of two types of birds on different backgrounds. It contains images of birds directly cut and pasted on real-life images of different landscapes. We aim at classifying $\mathcal{Y} = \{\text{waterbird, landbird}\}$, which is spuriously correlated with the background $\mathcal{A} = \{\text{water, land}\}$. The train/validation/test splits is followed from Sagawa et al. (2019).

**CelebA** (Liu et al., 2015): CelebA is an image dataset consisting of celebrities with two types of hair color and different genders. we aim at classifying celebrities' hair color $\mathcal{Y} = \{\text{blond, not blond}\}$, which is spuriously correlated with the gender $\mathcal{A} = \{\text{male, female}\}$. The train/validation/test splits is followed from Sagawa et al. (2019).

**MultiNLI** (Williams et al., 2018): MultiNLI is a natural language processing dataset, where each sample consists of two sentences and the second sentence is entailed by, neutral with, or contradicts the first sentence, with the presence or absence of negation words. We aim at classifying the sentence relationship $\mathcal{Y} = \{\text{entailment, neutral, contradiction}\}$, which is spuriously correlated with the negation words $\mathcal{A} = \{\text{negation, no negation}\}$. The train/validation/test splits is followed from Sagawa et al. (2019).

**CivilComments-WILDS** (Borkan et al., 2019; Koh et al., 2021) CivilComments-WILDS is a natural language processing dataset consisting of online comment that is toxic or non-toxic, with or without the mentions of certain demographic identities (male, female, White, Black, LGBTQ, Muslim, Christian, and other religions). We aim at classifying whether the sentence is toxic $\mathcal{Y} = \{\text{toxic, non-toxic}\}$, which is spuriously correlated with the mentions of (demographic) identities $\mathcal{A} = \{\text{identity, no identity}\}$. The train/validation/test splits is followed from Koh et al. (2021).

# B  TRAINING DETAILS

In this section, we provide details about the model selection and the hyperparameter tuning for different datasets. As claimed in the main text, in order to make a fair comparison with previous methods, we use the same pretrained models. Namely, we use ResNet-50 pretrained from Image-net weights for Waterbirds and CelebA, and pretrained BERT for MultiNLI and CivilComments. We use the `Pytorch` implementation for ResNet50 and the `HuggingFace` implementation for BERT. We tune BAM and JTT according to class difference and worst-group accuracies in the validation set in Stage 2.

Table 6: Hyperparameters tuned over 4 datasets.

| Dataset | Auxiliary coefficient ($\lambda$) | #Epochs in Stage 1 ($T$) | Upweight factor ($\mu$) |
|---|---|---|---|
| Waterbirds | $\{0.05, 0.5, 5\}$ | $\{10, 15, 20\}$ | $\{50, 100, 140\}$ |
| CelebA | $\{0.05, 0.5, 5\}$ | $\{1, 2\}$ | $\{50, 70, 100\}$ |
| MultiNLI | $\{0.05, 0.5, 5\}$ | $\{1, 2\}$ | $\{4, 5, 6\}$ |
| CivilComments | $\{0.05, 0.5, 5\}$ | $\{1, 2\}$ | $\{4, 5, 6\}$ |

In general, our setting follows closely from Liu et al. (2021), with some minor discrepancies. For the major hyperparameters, We tuned over $\lambda = \{0.05, 0.5, 5\}$, $T = \{1, 2, 10, 15, 60\}$ and $\mu = \{4, 5, 6, 50, 70, 100, 140\}$ for BAM. Despite the three choices for $\lambda$, we actually fix $\lambda = 0.5$ throughout our studies since it yields the best result and in fact, all three of $\lambda$'s yield very similar output under similar conditions. We note that BAM is fairly insensitive with the choice of $\lambda$. We tune over $T = \{1, 2\}$ and $\mu = \{4, 5, 6, 50, 70, 100, 140\}$ for JTT for fair comparisons. We tuned the major hyperparameters according to Table 6. More details are provided below:

**Waterbirds**  We use the learning rate 1e-5 and batch size 64 for two stages of training. We use the stochastic gradient descent (SGD) optimizer with momentum 0.9 throughout the training process. We use $\ell_2$ regularization 1 for Stage 2 *rebalanced training*. We apply the same above setting for both JTT and BAM. Notably, as illustrated by Figure 3, when tuned for the minimum absolute validation class difference, the curve may fluctuate abnormally after the first 30 epochs and it is clear that the

model is not learning anything useful. We tackle this problem by smoothing out abrupt changes in class difference (neglect the result if the difference between consecutive class differences is greater than 10%). The best result for JTT occurs when $T = 60$ and $\mu = 140$. The best result for BAM occurs when $T = 20$ and $\mu = 140$. We train for a total of 360 epochs

**CelebA**  We use the learning rate 1e-5 and batch size 128 for two stages of training. We use the stochastic gradient descent (SGD) optimizer with momentum 0.9 throughout the training process. We use $\ell_2$ regularization 0.1 for Stage 2 *rebalanced training*. We apply the same above setting for both JTT and BAM. The best result for JTT occurs when $T = 1$ and $\mu = 70$. The best result for BAM occurs when $T = 1$ and $\mu = 50$. We train for a total of 60 epochs.

**MultiNLI**  We use batch size 32 for two stages of training. We apply an initial learning rate of 2e-5 for Stage 1 and 1e-5 for Stage 2. We use the SGD optimizer without clipping for Stage 1 and the AdamW optimizer with clipping for Stage 2. We use $\ell_2$ regularization 0.1 for Stage 2 *rebalanced training*. We apply the same above setting for both JTT and BAM. The best result for JTT occurs when $T = 2$ and $\mu = 4$. The best result for BAM occurs when $T = 2$ and $\mu = 6$. We train for a total of 10 epochs.

**CivilComments-WILDS**  We use batch size 16 for two stages of training. We apply an initial learning rate of 2e-5 for Stage 1 and 1e-5 for Stage 2. We use the SGD optimizer without clipping for Stage 1 and the AdamW optimizer with clipping for Stage 2. We use $\ell_2$ regularization 0.01 for Stage 2 *rebalanced training*. We apply the same above setting for both JTT and BAM. The best result for JTT occurs when $T = 1$ and $\mu = 4$. The best result for BAM occurs when $T = 1$ and $\mu = 4$. We train for a total of 10 epochs.

## C  ADDITIONAL FIGURES AND STATISTICS

### C.1  SUPPLEMENTARY FIGURES

Figure 4 shows some supplementary plots to Figure 2, where from (1) and (2) it can be observed that using different upweight factors $\mu$ in Stage 2 will not change our findings that *One-M* on BAM performs better than *Two-M*, while *Two-M* on JTT performs better than *One-M*. In addition, Figure 5 shows that even in the case without any group annotations, BAM can still have a comparable result with JTT by stopping when achieving minimum class difference on the validation set.

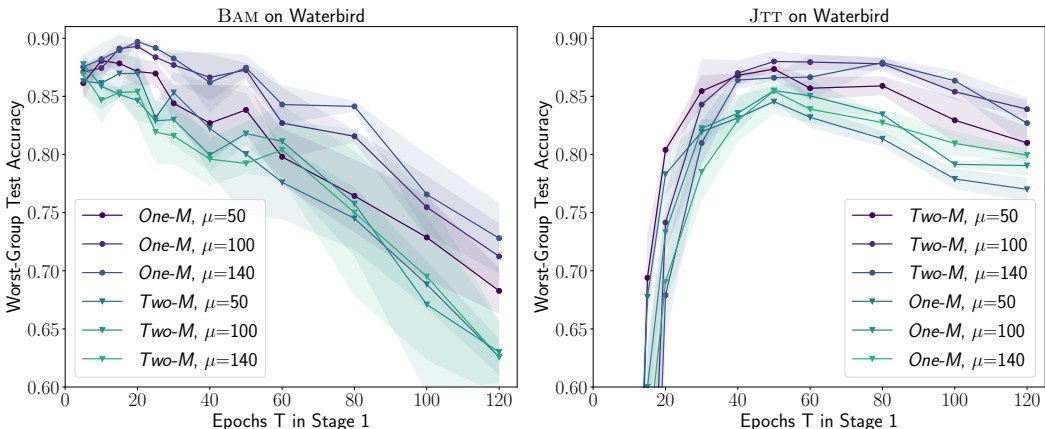

Figure 4: (1) Comparison of performance of BAM between training one model and two models through different upweight factor $\mu$ on Waterbirds. (2) Comparison of performance of JTT between training one model and two models through different upweight factor $\mu$ on Waterbirds.

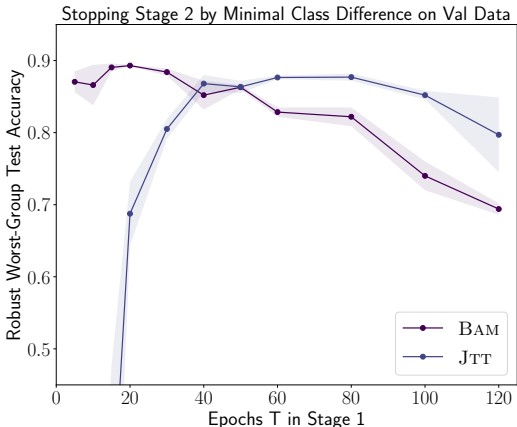

Figure 5: Comparison of performance of BAM against JTT by stopping at the minimum class difference on validation data by choosing their best set of hyperparameters separately. We can see BAM outperforms JTT with fewer epochs $T$ in Stage 1

## C.2 LESS VALIDATION SET

Table 7: Worst-group accuracy on Waterbirds with varying size of validation set with group annotations. BAM maintains high worst-group test accuracies even when tuned with very few numbers of group-annotated validation set. However, we note that with a fewer size of validation set with annotations, the performance is actually worse than when tuned for a full-size validation set without any annotations.

| Size of Annotated Validation Set | 100% | 20% | 10% | 5% |
|---|---|---|---|---|
| Worst-Group Acc.(%) | 89.8 | 89.1 | 88.4 | 86.2 |

# D REBUTTAL UPDATES

## D.1 VISUALIZATION OF AUXILIARY VARIABLES

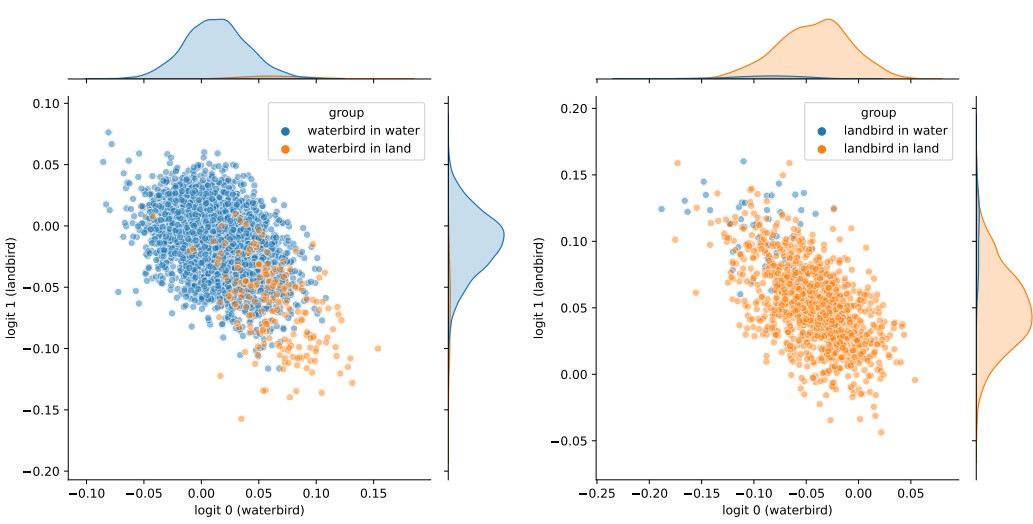

Figure 6: Distributions of the auxiliary variable w.r.t. the waterbird class (left) and landbird class(right) on the training set stopping at $T = 20$. We use two distinct colors to illustrate distributions of two groups in each class. The coordinates of data sample $i$ relative to the origin show the bias learned by the aux. variable.

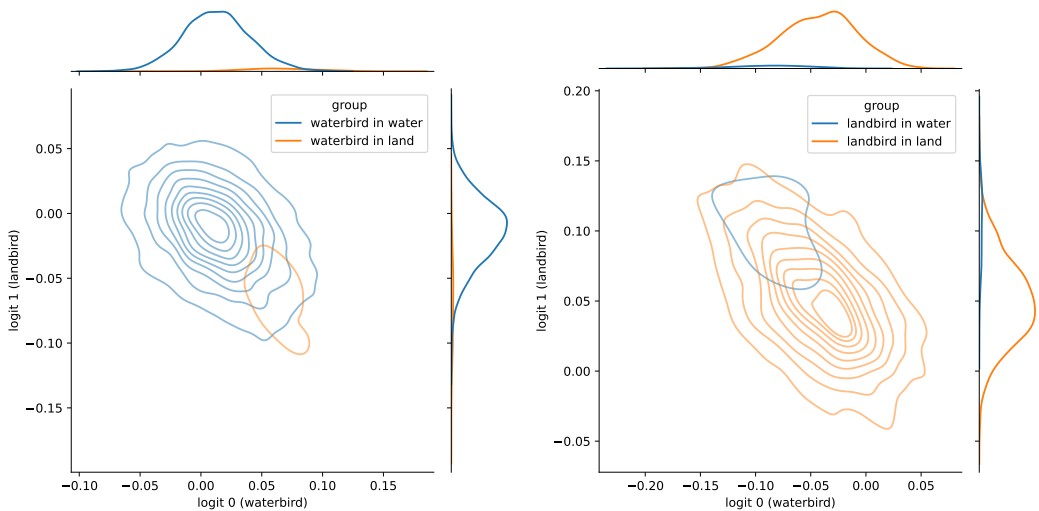

Figure 7: Corresponding KDE plot of Figure 6.

In Figure 6, we have two observations regarding the distribution of auxiliary variables on the training set. First, values of data samples in the majority and the minority group have clear distinctions for each class. This shows that the auxiliary variables differentiate between easy-to-learn and hard-to-learn examples. Second, auxiliary variables for majority group examples are in general closer to the origin, while those in minority groups tend to have larger (positive) logit values on the ground truth class they actually belong to and have smaller (negative) logit values on the class they do not belong to. The visualization shows that the auxiliary variables help with fitting hard-to-learn samples, which corroborates the intuition described in Section 4.1. We observe the same trend for any

$\lambda \in \{0.5, 5, 20, 50, 70, 100\}$ as well as any $T$ in Stage 1 in $\{20, 50, 100\}$, which strongly supports our claims that "adding auxiliary variables amplifies the bias toward easy-to-learn examples" in Section 4.1.

### D.2   HOW AUX. VARIABLE CHANGES AS TRAINING PROCEEDS

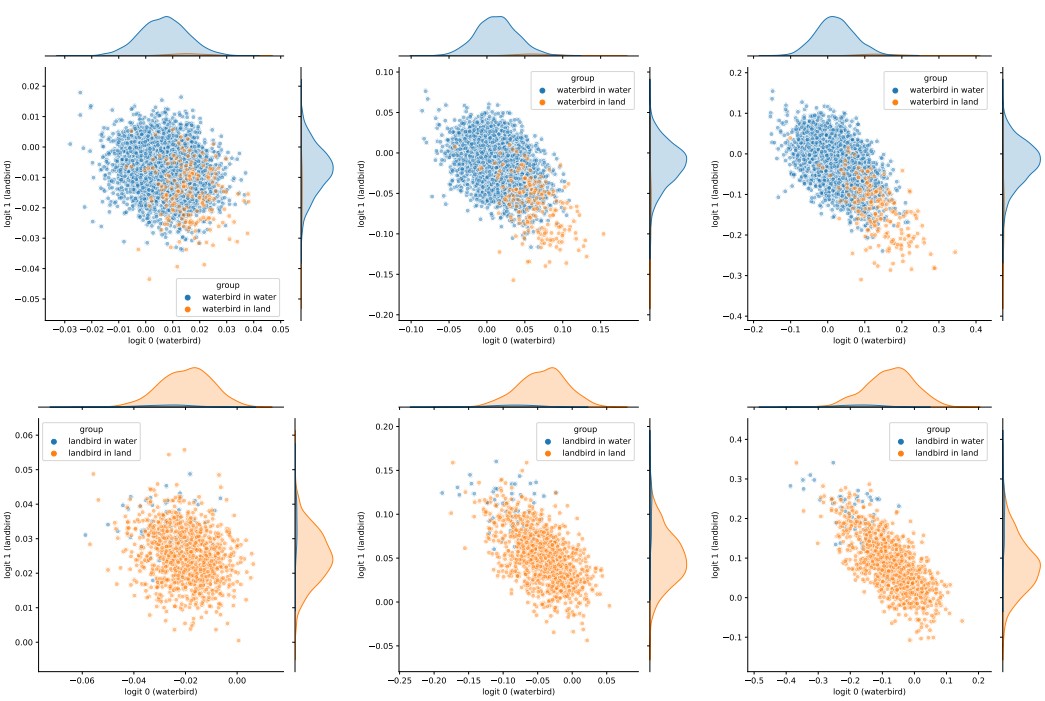

Figure 8: Distribution changes of the auxiliary variable w.r.t. the waterbird class (upper) and landbird class(lower) on the training set stopping at $T = 5$ (left), $T = 20$ (middle), and $T = 50$ (right).

In Figure 8, we observe that as training proceeds, the magnitudes of auxiliary variables get larger, and the minority group and majority group within each class can be easier to split apart.

### D.3   STABILITY OF $T$ AND ROBUSTNESS OF $\lambda$

First, we find that BAM is robust to the choice of $\lambda$, and below we present the best test accuracies obtained from a wide range of $\lambda$ using the waterbird dataset.



Table 8: Robustness of $\lambda$s

| $\lambda$ | Best worst group test accuracy |
|---|---|
| 0.5 | 89.9 |
| 10 | 89.8 |
| 30 | 89.7 |
| 35 | 90.2 |
| 40 | 89.5 |
| 50 | 89.5 |
| 70 | 88.8 |
| 100 | 88.6 |



The main benefit of using a larger $\lambda$ eliminates the need to carefully tune over $T$. As originally reported in Figure 2, we observe a degradation of performance as $T$ increases with a small choice of $\lambda$. Nevertheless, as we experiment with larger $\lambda$ that can yield comparable best worst-group accuracies, we find that it helps with preventing the original model from overfitting. Moreover,

Table 9: Robustness of worst group test accuracy with different $T$s

| $\lambda$ | $T$ | Worst group test accuracy |
|-----|-----|---------------------------|
| 50 | 80 | 88.6 |
| 50 | 100 | 88.2 |
| 50 | 120 | 86.9 |
| 70 | 80 | 88.8 |
| 70 | 100 | 88.8 |
| 70 | 120 | 88.5 |

there is no longer a need to carefully tune over $T$. Table 9 illustrates that even trained until full convergence, an appropriate choice of $\lambda$ can still guarantee a state-of-the-art performance.

