# OpenReview forum: "Bias Amplification Improves Worst-Group Accuracy without Group Information"
_ICLR.cc/2023/Conference — Submitted to ICLR 2023_

### Official Review · Reviewer_bEzt · 2022-10-18

**Confidence:** 4
**Correctness:** 3
**Technical Novelty And Significance:** 3
**Empirical Novelty And Significance:** 2
**Recommendation:** 5

**Clarity, Quality, Novelty And Reproducibility:**

## Clarity
Good.

The paper is well written and structured and easy to read. The idea is simple and effective.

## Quality
Fair.

The strengths are clear. There are however several major weaknesses in Methods and Experiments.

Please refer to the weaknesses part for a complete review.


## Novelty
Good.

Although the idea of introducing auxiliary variable during model training is proposed in the literature (Hu et al. 2020), adapting it for subgroup discovery and robustness seems to be novel and not explored before in the subpopulation field. It also gives insights on how to remove the dependency on the well-annotated validation set via a simple class-accuracy-difference criterion.

The literature review is comprehensive, and the paper is well positioned w.r.t. the literature.


## Reproducibility
Fair.

No code is provided along with the submission. The pseudo code is provided for the algorithm.

**Strength And Weaknesses:**

# Strengths
Overall the strengths of the paper are pretty clear. They can be summarized into the following points.

+ The idea of introducing learnable auxiliary variables with a bias amplification scheme for subgroup discovery and robustness seems to be novel and not explored before in this field.

+ The paper also introduced a simple yet effective method - class accuracy difference - to avoid the need of a balanced validation set with group annotations.

+ The author did a comprehensive experimental validation. They tested the BAM method on 4 popular subpopulation shift datasets, and explore the effects of the proposed 3 components via a thorough ablation study. The latter part is especially plausible as it makes the contribution of different components clearer.

+ The writing is overall good and easy to follow. The paper is well structured.

---

# Weaknesses
There are, however, several major weaknesses exist in the current paper.

## Method
- The authors claimed that "adding auxiliary variables amplifies the bias toward easy-to-learn examples". This is however not supported in the experiments.
1. What do the final learned variables look like after stage 1? It would be interesting and important to see the distribution of learned variables.
2. Are there any differences between that of minority groups versus majority groups? Is the bias really amplified toward easy-to-learn examples?

- The bias amplification scheme introduces an auxiliary variable for **each example** in the dataset. Moreover, by its definition, each variable has a size of $C$, which is the total number of classes. Therefore, the size of the auxiliary variables is propotional to the dataset size (total number of samples) as well as the total number of classes. It is doubtful that whether the method could scale to large-scale real-world datasets with large number of samples and fine-grained classification requirements (e.g., iNaturalist dataset has > 8,000 classes and tens of millions of images).

- Related to the above point, the idea of introducing auxiliary variables is plausible, but it also introduces potentially high computational costs. For example, ERM does not need any tracking or storing intermediate variables. I would like to see the actual computational cost w.r.t. time and memory costs. Wall-clock training time comparison as well as memory consumption to other baseline methods (e.g., ERM, JTT) would be good.

## Experiments
- Using squared loss instead of CE loss is interesting. The main intuition (motivation) comes from Hu et al. (2020). However, in Hu et al. (2020), using squared loss is reasonable as it recovers kernel ridge regression formulation. When it comes to the subpopulation setup in this paper, it is unclear why using squared loss is a good choice in the first place.
1. Can the authors explain, theoretically or hypothetically, that using squared loss is a good choice compared to CE loss in your setup?
2. The main change from CE to squared loss is using a soft regression loss instead of a hard classification loss. What about other (regression) losses, e.g., L1 or Huber loss? Is it actually the **squared loss** that leads to the improvement, or any of the aforementioned **regression losses** can lead to improvements? More comparisons are needed to confirm the argument.

- In Figure 2, regardless of whether One-M  or Two-M is used, the final performance of BAM degrades as $T$ gets large. This contrasts with the intuition, and also indicates that BAM training might not be stable and highly depends on the number of epochs used in stage 1.
1. Can the authors explain why performance of BAM degrades as $T$ gets larger?
2. Given its sensitivity to $T$, for a new dataset, how should one empirically choose the value of $T$? Does this indicate that BAM still needs a validation set to choose proper hyper-parameters?

---

# Additional questions

- Re-weighting is used for balanced training in stage 2. I wonder why not use data re-sampling (i.e., upsample the identified minority group or downsample the majority group). Is there a specific reason using only re-weighting?

- In Table 2, for the  CivilComments-WILDS dataset, the performances of BAM & JTT are exactly the same with or without annotations, but this does not hold for other datasets. I wonder is there a reason behind this? Intuitively with group annotations should improve the performance (at least for the worst group).

- Intuitively, the learned "bias" would be more stable when the model training is converged, which is however not the case (Figure 2). What if in stage 1, the network is trained longer enough? What would be the overfitting pattern? Will the model overfit to all training samples?

**Summary Of The Paper:**

This paper studies the problem of (in-distribution) subpopulation generalization when facing potential subgroups shift in the training data, without knowing the group annotations. The paper proposes a bias amplification scheme, which consists of two training stages. In the first stage, auxiliary variables are introduced for each input sample during model optimization, which are used to identify subgroups. Then the model is further trained using a re-weighting scheme based on the identified subpopulations of data points. It also proposes an early stopping method that does not need any subgroup annotations even in the validation set. Experiments on several datasets show that the method might be effective over several baselines on popular subpopulation shift datasets.

**Summary Of The Review:**

The paper studies an interesting yet important problem, model generalization when facing potential subgroups / subpopulations in the training data without knowing the group annotations.

The overall idea of leveraging a bias amplification scheme for subgroup discovery and robustness seems to be novel and not explored before in this field, which is plausible.

However, currently there are several drawbacks and weaknesses in terms of Methods and Exepriments, making me less confident on its stability, scalability, and working rationale.

In summary, my rating is borderline for now. I'm willing to change my score if my concerns/questions are well addressed.

---

> ### Author Response · Authors · 2022-11-15
> **Response to Reviewer bEzt (Part 2/2)**
>
> **Q1.4 Sensitivity of $T$**
>
> The experiment in Figure 2 was conducted for a relatively small $\lambda$ ($\lambda=0.5$), for which there is indeed an overfitting issue as $T$ gets larger. We have included additional experimental results for larger choices of $\lambda$ up to 100. We find that: (i) for larger $\lambda$, the performance degrades very little regardless of $T$, and (ii) the performance is stable across a wide range of $\lambda$.
>
> The following table shows that for larger $\lambda$, the worst-group accuracy is stable as $T$ gets larger.
>
> | $\\lambda$ | $T$ | Worst-group Acc.(%) |
> |-----------|-----|---------------------------|
> | 50        | 80  | 88.6                      |
> | 50        | 100 | 88.2                      |
> | 50        | 120 | 86.9                      |
> | 70        | 80  | 88.8                      |
> | 70        | 100 | 88.8                      |
> | 70        | 120 | 88.5                      |
>
> The following table shows that the worst-group accuracy is stable across a wide range of $\lambda$.
>
> | $\\lambda$ | Worst-group Acc.(%) |
> |-----------|--------------------------------|
> | 0.5       | 89.9                           |
> | 10        | 89.8                           |
> | 30        | 89.7                           |
> | 35        | 90.2                           |
> | 40        | 89.5                           |
> | 50        | 89.5                           |
> | 70        | 88.8                           |
> | 100       | 88.6                           |
>
> The above results show that BAM is robust with respect to the choice of $\lambda$, and for sufficiently large $\lambda$, early stopping in Stage 1 is no longer necessary.
>
> **Additional Question 1 - Reasons for using only re-weighting instead of data re-sampling**
>
> We indeed use upsampling in the implementation of Stage 2. We call it "re-weighting" following the terminology in the JTT paper. We will clarify this in the updated version.
>
> **Additional Question 2 - Reasons for the same performance on CivilComments-WILDS with or without annotations**
>
> This is not a mistake. Note that our proposed absolute class difference is simply a stopping criterion and does not change the nature of our training method. It is possible that minimum class difference identifies the same epoch as the highest worst-group validation accuracy, which turns out to be the case for CivilComments.
>
> **Additional Question 3 - Will the model overfit the training data if the network is trained longer enough?**
>
> As shown in our response to Q1.4, with an appropriate choice of $\lambda$, the learned "bias" is stable when training converges and there is no need of early stopping.

---

> ### Author Response · Authors · 2022-11-15
> **Response to Reviewer bEzt (Part 1/2)**
>
> Thank you for your valuable comments! We have addressed all your concerns and hope that you will consider raising the score.
>
> **Q1.1 Question on the distribution of the final learned Aux. Variables**
>
> Thank you for raising this great question. We have conducted additional experiments to study the distribution of the auxiliary variables $b_i$'s. We find that the auxiliary variables for minority group samples clearly have larger magnitudes than for majority group samples. This trend consistently appears for any $\lambda \in \\{0.5, 5, 20, 50, 70, 100\\}$ as well as any epoch number $T \in \\{20, 50, 100\\}$ in Stage 1. This exactly verifies our main intuition described in Section 4.1 that minority samples rely more on the auxiliary variables. Please see Figures 6 and 7 in Appendix D.1 for the results.
>
>
> **Q1.2 Concerns on Scalability of the Aux. Variables**
>
> It is possible to scale our method to very large datasets. In fact, we can store the auxiliary variables on the disk instead of in memory. During training, we only load the auxiliary variables corresponding to the current mini-batch, update these variables and save their new values on the disk. In this way, we only incur little additional memory and computation compared to plain ERM per iteration.
>
> We also observe that the algorithmic changes from JTT to BAM can lead to faster convergence, resulting in a shorter wall-clock time. For example, Figure 2 shows that the optimal performance of BAM is achieved at $T = 20$ compared with JTT's $T = 60$, resulting in 22min 42s shorter amount of time in Stage 1 using the exact same machine (11min 27s for BAM and 34min 9s for JTT).
>
> **Q1.3 Question on the loss functions**
>
> Thank you for your insightful comments on the loss function. We have conducted the following ablations on Waterbirds with the same hyperparameter settings reported in the paper: (1) L1-only (2) L1+AUX (3) Huber-only (4) Huber+AUX. We find that L1 loss and Huber loss both achieve competitive performance, as shown in the following table: (results are averaged over 3 runs)
>
> | Ablation                                    | Worst-group Acc.(\%) |
> |---------------------------------------------|------------------------------------------|
> | (1) Baseline (Cross-Entropy Loss)           | 86.4                                     |
> | (2) Baseline + Aux. Variable                |  88.7              |
> | (3) Baseline + Squared Loss                 | 88.8                                     |
> | (4) Baseline + Aux. Variable + Squared Loss | 89.7                                     |
> | (5) Baseline + L1 Loss                      | 88.8                                     |
> | (6) Baseline + Aux. Variable + L1 Loss      | 89.2                                     |
> | (7) Baseline + Huber Loss                   | 88.7                                     |
> | (8) Baseline + Aux. Variable + Huber Loss   | 89.2                                     |
>
> These results provide evidence that regression loss is crucial for improved worst-group performance, and the performance is better for the combined use of regression loss and auxiliary variables. We will add these results and discussion to the final version of the paper.
>
> Currently, we do not have a theoretical explanation of the advantage of regression loss over cross-entropy loss. We believe that this is a very interesting question for future studies. Even in the i.i.d. setting, it has been empirically observed that squared loss is competitive for classification tasks (e.g. "Evaluation of Neural Architectures Trained with Square Loss vs Cross-Entropy in Classification Tasks" by Hui and Belkin). There has been some initial theoretical explanation in high-dimensional linear models, e.g. "Classification vs regression in overparameterized regimes: Does the loss function matter?" by Muthukumar et al., from which some of the insights might transfer to deep learning.

---

> ### Author Response · Authors · 2022-11-28
> **We look forward to hearing your feedback!**
>
> Thanks for taking the time to share your feedback. We revised the manuscript and performed further analysis to address your concerns. Please see our inline response for more details. We look forward to hearing more about your thoughts and would be happy to answer more follow-up questions.

---

> > ### Comment · Reviewer_bEzt · 2022-12-11
> > **Response to Authors**
> >
> > First of all, I thank the authors for providing additional experiments and clarifications. I appreciate the efforts the authors made during the discussion phase.
> >
> > While some of the results indeed seem interesing, they still do not (directly) address my questions. For example, in the paper the authors claim that squared losses are essential; however, the new results using other regression losses seem to achieve similar benefits. They seem to highlight that it is the *regression* formulation that leads to the actual benefits over the CE loss. Similar issues happen also in the choices of resampling or reweighting. In short, which particualr method you choose does not matter; what matters is that you need to justify the choice. Currently I feel the rationales behind many design choices are rather empirical and "random", and not well justified.
> >
> > Considering the above as well as taking other reviewer's comments into account, I decide to keep my original rating. I do believe that the techniques and observations in the paper will have potential impacts for a broader audience in the field, but it will need extra careful revisions to incorporate the results and discussions with all reviewers, as well as more insights behind the observations.

---

> > > ### Author Response · Authors · 2022-12-11
> > > **Follow up from authors**
> > >
> > > We thank Reviewer bEzt for the reply. We are not sure why the reviewer said "many design choices are rather empirical and 'random', and not well justified." We clarify further below and are happy to answer any additional questions.
> > >
> > > - Regression losses: We will change our wording to regression losses instead of squared loss only, and will add all the results for L1 and Huber losses to the paper. We do not think this undermines the main message and results of this paper in any way, but rather reveals an interesting distinction between regression losses and CE loss. Understanding this difference is a deep question and is beyond the scope of this paper. Given the increasing popularity of using regression losses (especially squared loss) in classification problems, we do not think this should be viewed a random design choice.
> > >
> > > - Moreover, the key component of our method, auxiliary variable, has been justified with convincing evidence verifying its bias amplification effect by showing a clear difference between majority and minority samples. We also showed the robustness of our algorithm to hyperparameters including $\lambda$ and $T$, and that it works well under varying imbalance ratios. Overall, our algorithm is simple and clearly motivated, achieves state-of-the-art performance on standard benchmarks, and is robust to hyperparameter choices and dataset imbalance.
> > >
> > > - Resampling or reweighting: This is completely due to a confusion of terminology in the literature -- oftentimes they are used interchangeably. Both our paper and the JTT paper use resampling, not reweighting. As mentioned in the paper, we did not alter Stage 2 of JTT because our focus was on improving Stage 1. We apologize for this confusion.
> > >
> > > Please let us know if there are any other concerns.

---

### Official Review · Reviewer_3fdp · 2022-10-23

**Confidence:** 4
**Correctness:** 3
**Technical Novelty And Significance:** 2
**Empirical Novelty And Significance:** 2
**Recommendation:** 5

**Clarity, Quality, Novelty And Reproducibility:**

- The overall writing is easy to read.
- The main part of this work I think is about auxiliary variable, but there is no analysis on this, which weakens its novelty.
- There is no source codes, but the pseudo-codes are clear and easy to follow.

**Strength And Weaknesses:**

### Strength
- The paper provides performance improvement in most of datasets.
- The authors also introduce early stopping strategy without group annotations to ensure the unsupervised setting.
- The writing is easy to follow.

### Weakness
- The main idea is partially overlapped with LfF (stage 1) and JTT (stage 2), and the performance improvement is not significant compared to theirs. The auxiliary variable aims to make hard examples to be learnt harder, which has similar objective to generalized cross entropy of LfF. What is the advantage of auxiliary variable, compared to GCE of LfF?
- Why the auxiliary variable is instance-wise, rather than the common one for all samples? If I correctly understand, this is not realistic if there are millions or billions of training images. Also, how does the auxiliary variable changes as training progresses?
- There are some missing comparisons for unsupervised debiasing methods [LWBC, BPA, CVaR DRO]. Especially, [LWBC] also employs a biased committee to learn a final debiasing classifier.

[LWBC] Learning Debiased Classifier with Biased Committee, NeurIPS 2022

[CVaR DRO] Large-scale methods for distributionally robust optimization, NeurIPS 2020

[BPA] Unsupervised learning of debiased representations with pseudo-attributes, CVPR 2022

### Minor questions
- Why One-M is better than two-M only after bias amplification? There is no explanation.
- The results of BAM/BAM+class diff and JTT/JTT+class diff on CivilComments are the same. Is it correct?



**Summary Of The Paper:**

Several group robust optimization have achieved remarkable performance but they require group annotations, which limits the practicality. To alleviate this limitation, the authors focus on the scenario where group annotations are not or partially available on a validation set. The proposed method BAM consists of two stages; 1) model is trained using a bias amplification scheme with an auxiliary variable and 2) upweight the samples that are misclassified in 1).

**Summary Of The Review:**

- The authors proposed a two-stage methods for learning debiased representations, but however, I don’t think this method is sufficiently novel compared to the existing approaches. For a stronger work, I think it needs to conduct more detailed and thorough analysis about auxiliary variable for debiasing.

---

> ### Author Response · Authors · 2022-11-15
> **Response to reviewer 3fdp**
>
> Thank you for your valuable comments! We have addressed all your concerns and hope that you will consider raising the score.
> Please find our replies below.
>
> **Q2.1 Questions on (1) Novelty (2) The advantage of Aux. Variables over GCE of LfF (3) Performance of BAM against LfF and JTT**
>
> While we agree that the main idea is partially overlapped with LfF and JTT, there are several key differences between BAM and them. Note that the other two reviewers both appreciate the novelty of our method. Below we highlight some of the key differences.
>
> First, we'd like to emphasize that the performance improvement of BAM over LfF and JTT is consistent in all the benchmark datasets we tested on, and is by a large margin in some of them, as summarized in the table below:
>
> | Method | Waterbird | CelebA | MultiNLI | CivilComments-WILDS |
> |--------|-----------|--------|----------|---------------------|
> | LfF    | 78.0\%    | 77.2\% | 70.2\%   | 58.8\%              |
> | JTT    | 86.8\%    | 78.7\% | 68.1\%   | 77.7\%              |
> | BAM    | 89.7\%    | 83.5\% | 71.5\%   | 79.3\%              |
>
> Table 1: Worst-group Accuracy of LfF, JTT, and BAM
>
> Regarding the comparison between auxiliary variables and the GCE loss in LfF, they focus on different components and may not be directly comparable: GCE is a loss function, while auxiliary variables directly manipulate the model outputs. It would be more appropriate to compare GCE with cross-entropy loss and squared loss. In fact, we conducted comprehensive experiments using GCE, and found that it has worse accuracy and longer convergence speed than squared loss. We will add these discussions to the final version of the paper.
>
> Another important element in BAM is "one-M'' (continued training) which differentiates it from LfF and JTT which train two separate models. This is effective when combined with bias amplification in Stage 1 (see the ablation studies in Section 5.3).
>
> Additionally, in order to demonstrate the effectiveness of bias amplification in Stage 1, we only use the most naive Stage 2 (directly adopted from JTT) for the purpose of fair comparison. We believe it is possible to use other more complicated methods in Stage 2 for further improvement.
>
> **Q2.2 (1)Why the auxiliary variable is instance-wise? (2) Scalability of the Aux. Variables. (3)How does the auxiliary variable changes as training progresses?**
>
> - **To the Question of why the aux. variable is instance-wise:** It is very important that the auxiliary variables are instance-wise. As mentioned in Section 4.1, we expect hard-to-learn examples to rely more on their auxiliary variables than easy-to-learn examples, and therefore we need instance-wise auxiliary variables to differentiate between hard-to-learn and easy-to-learn examples. Our additional experiments in Appendix D.1 show that the learned auxiliary variables indeed have larger magnitudes for minority group examples than majority ones.
>
> - **To the Question of scalability of aux. variable:** It is possible to scale our method to very large datasets. In fact, we can store the auxiliary variables on the disk instead of in memory. During training, we only load the auxiliary variables corresponding to the current mini-batch, update these variables and save their new values on the disk. In this way, we only incur little additional memory and computation compared to plain ERM per iteration.
>
> - **To the Question of how does auxiliary variable changes as training progresses?** Thank you for the great question. We have conducted a detailed analysis of the distribution of the auxiliary variables during training. Please see Figure 8 in Appendix D.2. Our main findings are: (i) there is a clear distinction between auxiliary variables for minority group examples and majority group examples, the former having larger magnitudes; (ii) as training progresses, this distinction becomes more and more clear. This verifies our intuition described in Section 4.1.
>
> **Q2.3 Citations**
>
> Thank you for pointing these papers out. We have cited and discussed them in the related work section. The methods in these papers are significantly different from ours, and they either achieve worse performance or were tested on different datasets.
>
> **Minor question 1: why One-M better than Two-M only after bias amplification?**
>
> One possible explanation is that bias amplification together with One-M provides a better curriculum for the model, so that it focuses more on easy-to-learn examples in Stage 1 and hard-to-learn examples in Stage 2. We think it will be an interesting future direction to formalize this intuition.
>
> **Minor question 2: CivilComments ClassDiff**
>
> This is correct. Note that our proposed absolute class difference is simply a stopping criterion and does not change the nature of our training method. It is possible that minimum class difference identifies the same epoch as the highest worst-group validation accuracy, which turns out to be the case for CivilComments.

---

> ### Author Response · Authors · 2022-11-28
> **We look forward to hearing your feedback!**
>
> Thanks for taking the time to share your feedback. We revised the manuscript and performed further analysis to address your concerns. Please see our inline response for more details. We look forward to hearing more about your thoughts and would be happy to answer more follow-up questions.

---

### Official Review · Reviewer_UsCc · 2022-10-25

**Confidence:** 4
**Correctness:** 3
**Technical Novelty And Significance:** 3
**Empirical Novelty And Significance:** 3
**Recommendation:** 6

**Clarity, Quality, Novelty And Reproducibility:**

The paper is well written and clearly presented. Connections to related works are explained, motivating the proposed method well. The contribution is novel, as one of the few that explicitly links worst-group improvement literature with learning with label noise (LLN). The method should be easily reproducible as the algorithm is explained in detail.

**Strength And Weaknesses:**

Strengths:

1. The paper is one of the few that explicitly links the worst-group improvement literature with learning with label noise (LLN). A particular LLN model (Hu et al. 2020) is used to replace stage 1 of JTT. I think the idea can be quite interesting for many readers to further explore the links between the two research areas.
2. The paper explores worst-group improvement without any group annotations at all which is an understudied problem in the literature.
3. Abundant ablations are performed to support the proposed model. Some can be quite interesting to other readers, such as One-M vs Two-M (Fig 2).

Weaknesses: The method is clearly presented, and I only have some comments/clarification questions.

1. I find it a bit hard to understand the motivation for the ClassDiff validation metric for model selection. Why pairwise class perf diff should work? For example, is it  somehow related to worst-class accuracy (as opposed to worst-group that requires annotation)? How robust is it if one varies class imbalance or (group,class)-imbalance (as a tuple) in the val set? Some intuitions/ablations can help.
2. Is lambda in Eq. 1 fixed to 0.5 instead of tuned as a hparam (as said in Sec 5.3, but not sure how it's done in Sec 5.1/2?)
3. The paper discussed, and compared empirically through ablation, the choice of MSE vs CE as loss function in Stage 1. In fact, a relatively well known robust loss for LLN is simple MAE loss (https://arxiv.org/pdf/1712.09482.pdf). Given its implementation simplicity, it would be interesting to see how it compares to MSE/CE to suit this task.
4. As a minor point, despite being the an interesting and well-motivated idea to use a noise-robust bias amplifying model to replace JTT's stage 1 model, I still think the contribution can be much stronger if more LLN models are tested (and of course significantly more amount of work needed). Or at least I think it'd be good to make the point clear from the beginning that in fact many other LLN models (e.g. curriculum learning, robust losses, label transition) can be candidates to identify possibly group-conflicting training samples. (Perhaps another LLN model may suit the task even better with better performance.)
5. As common evaluation criteria of LLN models, one could also report the clean/noisy ratio (i.e. identified clean/noisy samples in the training set) to demonstrate how well the method works. A similar idea could be employed to show how well bias amplifying model compares to a simple early-stopped ERM-trained model (JTT) in terms of identifying group-conflicting training examples in another experiment/ablation.

**Summary Of The Paper:**

The paper proposes a two-stage training approach for improving worst-group performance, BAM (Bias Amplication), where in the first stage a bias amplified model is trained (Hu et al. 2020) to replace that of JTT, and the second stage still follows JTT. The paper also proposes a validation metric that works well without any group annotation on the validation set. Empirical results are reported and abundant ablations are performed to support the model.

**Summary Of The Review:**

The paper is one of the few that explicitly links the worst-group improvement literature with learning with label noise (LLN), and explores another understudied scenario where no group annotations are available at all. The paper can be improved if some comments above can be addressed.

---

> ### Author Response · Authors · 2022-11-15
> **Response to Reviewer UsCc**
>
> We thank the reviewer for reviewing our work and appreciate the positive feedback and questions. We address the questions below and hope that the reviewer will consider raising the score.
>
> **Q3.1 Motivation for ClassDiff**
>
> The main motivation for ClassDiff is that we expect the worst-group accuracy to be (near) highest when all group accuracies are similar. A weaker condition is that the class accuracies are similar, which can be estimated by ClassDiff without access to any group annotations. We found that this is an effective heuristic in all the benchmark datasets we tested. We agree that it will be valuable to do further ablations with respect to different imbalance ratios, and plan to include this in the final version.
>
> **Q3.2 Choice of $\lambda$**
>
> In the original submission, we tuned $\lambda$ in $\\{0.05, 0.5, 5\\}$ as mentioned in Appendix B, but we fixed $\lambda=0.5$ in all the reported results since we found that different choices of $\lambda$ have little effect on the final performance. We have conducted additional experiments in Waterbirds for a much wider range of $\lambda$ (0.5, 10, 30, 35, 40, 50, 70, 100) in Appendix D.3 and found that our method is still robust to the choice of $\lambda$.
>
> **Q3.3 Loss function**
>
> Thank you for pointing out MAE loss. We have conducted the following ablations on Waterbirds with the same hyperparameter settings reported in the paper: (1) L1-only (2) L1+AUX (3) Huber-only (4) Huber+AUX. We find that L1 loss and Huber loss both achieve competitive performance, as shown in the following table: (results are averaged over 3 runs)
>
> | Ablation                                    | Worst-group Acc.(\%) |
> |---------------------------------------------|----------------------|
> | (1) Baseline (Cross-Entropy Loss)           | 86.4                 |
> | (2) Baseline + Aux. Variable                | 88.7                 |
> | (3) Baseline + Squared Loss                 | 88.8                 |
> | (4) Baseline + Aux. Variable + Squared Loss | 89.7                 |
> | (5) Baseline + L1 Loss                      | 88.8                 |
> | (6) Baseline + Aux. Variable + L1 Loss      | 89.2                 |
> | (7) Baseline + Huber Loss                   | 88.7                 |
> | (8) Baseline + Aux. Variable + Huber Loss   | 89.2                 |
>
>
> These results provide evidence that regression loss is crucial for improved worst-group performance, and the performance is better for the combined use of regression loss and auxiliary variables. We will add these results and discussion to the final version of the paper.
>
> **Q3.4 \& 3.5 LLN models \& clean/noisy Ratio**
>
> We appreciate your suggestion of considering more LLN models and making more connections to the LLN literature. Due to the significantly more amount of work involved (as you mentioned), further exploration is beyond the scope of this paper. We strongly agree that this is a promising direction for future work, and will add more discussions around this in the introduction and conclusion of the paper.

---

> ### Author Response · Authors · 2022-11-28
> **We look forward to hearing your feedback!**
>
> Thanks for taking the time to share your feedback. We revised the manuscript and performed further analysis to address your concerns. Please see our inline response for more details. We look forward to hearing more about your thoughts and would be happy to answer more follow-up questions.

---

> ### Author Response · Authors · 2022-12-10
> **Updated Response for Q3.1 & Q3.5**
>
> We conducted additional experiments on two datasets to verify the robustness of BAM by changing the relative group sizes and class sizes: Colored-MNIST and Controlled-Waterbirds. On Colored-MNIST, we set up the experiment with a total dataset size of 20000, dataset splits of \{0.6, 0.2, 0.2\}, $\lambda$ of 50, tuned over $T \in$ \{20, 50\} and $\mu \in$ \{5, 10, 20, 50\}. On controlled waterbird, we fixed $\lambda$ of 50, $T$ of 100, tuned over $\mu \in$ \{10, 50, 70, 100, 120, 150\}. **The following tables illustrate that BAM is robust to varying group and class ratios.** All results are averaged over three random experiments. In addition, we observed on every single experiment of both these two new established datasets that **the class difference is negatively correlated with highest robust test accuracy**, regardless of their group sizes and class sizes. More illustrative graphs of this correlation w.r.t. different parameters to construct the datasets will be added to the Appendix in the camera ready version.
>
> **Varying ratio of group sizes on Colored-MNIST. Two classes are balanced:**
>
> | Minority group size : Majority group size | Worst-group Acc.(\%) |
> |-------------------------------------------|----------------------|
> | 0.05                                      | 93.1                 |
> | 0.1                                       | 94.9                 |
> | 0.2                                       | 97.4                 |
> | 0.3                                       | 97.3                 |
>
> **Varying ratio of class sizes on Colored-MNIST. Fix Minority group size : Majority group size as 0.1.**
>
> | Majority class size : Minority class size | Worst-group Acc.(\%) |
> |-------------------------------------------|----------------------|
> | 1.0                                       | 94.9                 |
> | 1.5                                       | 96.4                 |
> | 2.0                                       | 95.3                 |
> | 2.5                                       | 95.9                 |
>
> **Varying ratio of group sizes on Controlled-Waterbirds. Two classes are balanced. Fix either of the majority group sizes as 1800.**
>
> | Majority class size : Minority class size | Worst-group Acc.(\%) |
> |-------------------------------------------|----------------------|
> | 1/9                                       | 87.8$\pm$0.03        |
> | 2/9                                       | 83.0$\pm$0.02        |
> | 3/9                                       | 83.5$\pm$0.01        |
> | 4/9                                       | 86.7$\pm$0.01        |
>
> **Varying ratio of class sizes on Controlled-Waterbirds. Fix the smallest minority group size as 200, Minority group size : Majority group size as 1:7.**
>
> | Majority class size : Minority class size | Worst-group Acc.(\%) |
> |-------------------------------------------|----------------------|
> | 1.0                                       | 86.1$\pm$0.04        |
> | 2.0                                       | 83.1$\pm$0.03        |
> | 3.0                                       | 88.2$\pm$0.01        |

---

### Official Review · Reviewer_CiNy · 2022-12-12

**Confidence:** 4
**Correctness:** 3
**Technical Novelty And Significance:** 2
**Empirical Novelty And Significance:** 2
**Recommendation:** 5

**Clarity, Quality, Novelty And Reproducibility:**

**Clarity and Quality**
The paper is very clear. All the related research is properly described.

**Novelty**
On the technical side, the paper does not offer significant novel contributions. Similar to JTT, it is a two-stage approach that proceeds by first identifying the error set and later up weights the samples based on the error set.

**Reproducibility**
Pseudo-code is provided, which can help in assessing reproducibility.


**Strength And Weaknesses:**

**Strengths**

A strength in the paper is the introduction of the ClassDiff metric. The authors verify the efficacy of the metric in being used as a proxy for worst-off group accuracy in their experiments. Prior works like JTT and EIIL use group annotations for hyper-parameter tuning, however, obtaining group annotations is often an expensive task. For this reason, it is essential for a method to avoid any group annotations during training or validation.

The ClassDiff metric is analogous to a fairness measure called demographic parity. Additionally, Worst-off group accuracies is captured by the fairness measure called Rawlsian Criterion. The experiments demonstrate that on the datasets used in the paper demographic parity can be used in place of the Rawlsian Criterion for hyper-parameter tuning.

The auxiliary variable technique effectively identifies error sets and improves the worst-off group accuracy. This is demonstrated via experiments where the overall accuracy and worst-off group accuracy are comparable to the baselines.

The paper provides a good list of ablation experiments. The most helpful ones are the One-M / Two-M model comparisons and the experiments identifying inverse relationship between ClassDiff and Worst-off Group Accuracy.

**Weaknesses**

There is an inherent trade-off between the average group accuracy and the worst-off group accuracy. The experiments demonstrate that the proposed method provides a small drop in the overall accuracy at the cost of improving the worst-off group accuracy. It maybe required to observe the pareto-frontier of the proposed method and the other baselines to asses if the proposed method provides better metrics for a wide range of hyper-parameters. At the moment, the experiments suggest that the proposed method is comparable (and not strictly better) than the baselines when evaluated for both average accuracy and worst-offf group accuracy.

While the paper suggests that ClassDiff (or the demographic parity) measure can be used as a proxy for Worst-off group accuracy (or the Rawlsian Criterion), it is unclear if such a proxy would work always. A more thorough study, either through theoretical analysis or experimental evidence, is needed to identify scenarios in which ClassDiff is to be (and is NOT to be) used.

The method proposed in the paper, similar to JTT, is a two-stage approach. Any two-stage approach requires 2X training time relative to ERM. It is a weakness of the method that it requires 2X training time relative to ERM.

**Summary Of The Paper:**

I am adding an additional review to aid AC in making a decision for the paper.

The paper proposes a method to improve the worst-off group accuracies in a dataset when the group annotations are only available during validation. Prior research either uses group annotations during training (such as GroupDRO) or trains in an unsupervised manner as a two-stage method (such as JTT).

A two-stage method is proposed in the paper where an error set of hard-to-learn examples is identified in stage one. These examples are devoid of any spurious correlations and are identified via learnable auxiliary variables and a squared loss. In stage-two, the sample misclassified by stage-one are upweighted.

For tuning hyper-parameters, a metric called ClassDiff is introduced. This metric captures the pair-wise class difference between the accuracies. Experiments show it to be a good proxy for worst-off group accuracy.

Experimental results show an improvement over the worst-off group accuracy in four benchmark datasets.


**Summary Of The Review:**

The paper proposes a two-stage approach for improving the worst-off group accuracies. The method bears similarities to JTT, as both the approaches first identify and error set and later upweight the samples based on the error set. On the experimental side, the method is comparable to the baselines when evaluated for both average group accuracy and worst-off group accuracy. While it is interesting to see an unsupervised metric, ClassDiff, being for hyper-parameter tuning, it is unclear when such a metric is applicable and when should one avoid using ClassDIff. Based on these reasons, I am leaning towards a reject.

---

> ### Author Response · Authors · 2022-12-13
> **Reponse to Reviewer CiNy**
>
> We thank the reviewer for providing the feedback. We provide our responses below.
>
> **Q4.1 Average accuracy and worst-group accuracy tradeoff**
>
> We are puzzled by this concern about the tradeoff between average and worst-group accuracies. The focus of our paper, as well as a large body of papers in the area of spurious correlation and group robustness, is **exclusively on the worst-group accuracy**. It is not clear whether we should even care about the average accuracy in presence of spurious correlations. We mention explicitly on Page 7 (Section 5.2) that a moderate drop in average accuracy is consistent with previous approaches like JTT (Liu et al. 2021) and GroupDRO (Sagawa et al. 2019), both of which have been widely accepted by the community and received more than 100 citations. Requiring high worst-group accuracy and no loss in average accuracy moves the goalposts for a whole line of research.
>
> **Q4.2 ClassDiff**
>
> The reviewer mentioned that "a more thorough study, either through theoretical analysis or experimental evidence, is needed to identify scenarios in which ClassDiff is to be (and is NOT to be) used." We would like to emphasize that we did abundant experiments to evaluate ClassDiff and were not able to find any natural problem instance on which ClassDiff fails. Figure 3 in our paper shows the relation between absolute validation class difference and worst-group accuracy on all four benchmarks (Waterbirds, CelebA, CivilComments-WILDS, MultiNLI) used extensively in the literature. Moreover, we have done additional experiments on two more datasets, Colored-MNIST and Controlled-Waterbirds. On these datasets, we vary the dataset sizes, class size ratios, and group size ratios over a large range. **In every single setting we tested**, we observe similar negative correlations as in Figure 3. These findings suggest that ClassDiff could be generally useful whenever no group annotation is available, as it is robust across different datasets, varying dataset sizes, and imbalance characteristics.
>
> **Q4.3 Training time comparison with ERM**
>
> We would like to remind the reviewer that BAM outperforms ERM by an average of 36\% in worst-group accuracy on four benchmark datasets. We do not think it is fair to compare the training time between BAM and ERM given such a significant performance gap. In addition, as mentioned in our response to Reviewer bEzt, compared with JTT, BAM only takes $1/3$ time in Stage 1 due to much faster convergence.
>
> **Q4.4 Similarity with JTT (novelty)**
>
> Our method is sufficiently different from JTT due to the use of trainable auxiliary variables and changing from Two-M to One-M. The improved performance is consistent in all the settings we tested and significant in some of them. Note that both Reviewer bEzt and Reviewer UsCc appreciate the novelty of our method.

---

### Author Response · Authors · 2022-11-15
**General Response and Summary of Revision**

Dear Reviewers:

Thank you all for your thoughtful and constructive reviews. We have addressed all the questions and concerns in the response to each reviewer. Based on your feedback, we have conducted additional experiments regarding (i) the distribution of auxiliary variables and (ii) the stability of $T$ (number of epochs in Stage 1) and robustness of $\lambda$ (parameter for auxiliary variables). These results significantly strengthen our paper by solidifying the underlying intuition of auxiliary variables and verifying its robustness over hyperparameter choices. Please see the newly added Appendix D for the results.

Here, we first summarize some of the common strengths agreed upon by the reviewers:

- The majority of reviewers agree that our proposed method is novel and well-motivated from existing literature.
- All reviewers praise our proposed early stop criterion --- ClassDiff, which ensures competitive performances in fully unsupervised settings.
- The majority of reviewers agree that our experimental evaluations and ablation studies are conducted thoroughly and some ideas might be enlightening to other domains.
- All reviewers comment that our paper is well-written and easy to follow.

We emphasize a few key points from our responses and additional experiments:

- One of the major concerns of Reviewers bEzt and 3fdp is the scalability of auxiliary variables. In fact, it is possible to scale our method to very large datasets. We can save the auxiliary variables on the disk and load them in mini-batches, just like how we typically load a dataset. (See Q1.2 and Q2.2 for detailed response).
- We observe a clear distinction in the distribution of the auxiliary variables between the majority and minority groups, which supports our statement in the paper that "bias is amplified towards easy-to-learn  examples.'' (See Q1.1 and Appendix D.1)
- We show that BAM is robust to the choice of $\lambda$, and for large enough $\lambda$, the performance of BAM no longer deteriorates when trained until convergence. (See Appendix D.1 and Q1.4)
- Thanks to the suggestion from Reviewers bEzt and UsCc, we explored the use of other regression losses (L1 \& Huber) in Stage 1, and observed that regression loss is indeed an important factor contributing to the performance boost (Q1.3 and Q3.3).

---

### Author Response · Authors · 2022-12-10
**Additional General Response**

We have conducted additional experiments with varying group/class imbalance ratios on Colored-MNIST and Controlled-Waterbirds. On Colored-MNIST, we set up the experiment with a total dataset size of 20000, dataset split of $\\{ 0.6, 0.2, 0.2\\}$, $\lambda$ of 50, tuned over $T \in \\{ 20, 50\\}$ and $\mu \in \\{5, 10, 20, 50\\}$. On controlled waterbird, we fixed $\lambda = 50, T = 100$ and tuned over $\mu \in \\{ 10, 50, 70, 100, 120, 150\\}$. We change the group/class imbalance ratios respectively. The experimental results further confirm the **robustness of BAM under different degrees of imbalances** (See Tables on "Updated response for Q3.1 & Q3.5").  Additionally, the classDiff consistently yields competitive results compared to the previous stopping criteria of using the highest validation worst-group accuracies under all scenarios we tested.

---

### Decision · Program_Chairs · 2023-01-20

**Decision:**

Reject

**Justification For Why Not Higher Score:**

see above

**Justification For Why Not Lower Score:**

see above

**Metareview: Summary, Strengths And Weaknesses:**


There was an extensive discussion with the reviewers about this paper but all reviewers believed that the paper was not ready for publication in its current form.  Overall, it was felt that the paper proposes a very heuristic approach to improve a worst-group accuracy measure in a way that does not require group information even in the validation data. While this is an admirable goal, the method given is a simple two-stage heuristic that is evaluated on data, but there are no analyses as to if the heuristic should be expected to well in general, or if it is the case that it does ok on the data sets that were tested. Add to this that the idea of two-stages is itself not new, and not to mention that boosting algorithms from the 1990s essentially do a multi-stage re-waiting of later stage classifiers based on the error made by earlier stage classifiers (and boosting has extensive theoretical analysis) renders this paper fairly weak on the novelty front. The ClassDiff measure is simply the average difference of accuracies of the different classes, and it is claimed to be a surrogate for the worse group accuracy, but in general averages need not be good estimates of minimums, which is in fact the reason for robust analysis in the first place, so even if empirically you find that ClassDiff correlates well with worst group accuracy does not mean that it holds in general, or should hold on some new scenario.

Lastly, in my own reading of the paper, I believe there is a technical error right after equation (1). You say that $b_i \in \mathbb R^C$ which means that $b_i$ for each $i$ may be an arbitrary real-valued vector of length $C$, unbounded in magnitude and arbitrary in sign. Then for any non-zero $\lambda$, we can set $b_i(j)$ either to $1/\lambda$ or to 0 (depending on $e_y$) and get zero loss even when $f_\theta(x_i) = 0$, meaning that the $b_i$ can essentially memorize the labels and the model does not need to train at all. Since this is a squared 2-norm error, it would be a trivial solution for optimization to achieve. Unless you place some sort of constraints on the $b_i$ while training, then the loss can trivially be zero without training the model at all. Presumably this is an oversight in the presentation, but this clearly needs to be fixed in the paper.

**Summary Of Ac-Reviewer Meeting:**

see above